# The value of visualization in improving compound flood hazard communication: A complementary perspective through a Euclidean Geometry lens

Soheil Radfar [1,2*], Georgios Boumis [1,2], Hamed R. Moftakhari [1,2], Wanyun Shao[3], Larisa Lee[4], Alison N. Rellinger[4,5]

[1] Center for Complex Hydrosystems Research, Tuscaloosa, AL, USA
[2] Department of Civil, Construction, and Environmental Engineering, University of Alabama, Tuscaloosa, AL, USA
[3] Department of Geography & the Environment, University of Alabama, Tuscaloosa, AL, USA
[4] Coastal Research and Extension Center, Mississippi State University, Biloxi, MS, USA
[5] Mississippi-Alabama Sea Grant Consortium, Ocean Springs, MS, USA

*Correspondence to*: Soheil Radfar (sradfar@ua.edu)

**Abstract.** Compound flooding, caused by the sequence/co-occurrence of flood drivers (i.e. river discharge and elevated sea level) can lead to devastating consequences for society. Weak and insufficient progress toward sustainable development and disaster risk reduction are likely to exacerbate the catastrophic impacts of these events on vulnerable communities. For this reason, it is indispensable to develop new perspectives on evaluating compound flooding dependence and communicating the associated hazards to meet UN Sustainable Development Goals (SDGs) related to climate action, sustainable cities, and sustainable coastal communities. The first step in examining bivariate dependence is to plot the data in the variable space, i.e., visualizing a scatterplot, where each axis represents a variable of interest, then computing a form of correlation between them. This paper introduces the Angles method, based on Euclidean geometry of the so-called "*subject space*," as a complementary visualization approach specifically designed for communicating the dependence structure of compound flooding drivers to diverse end users. Here, we evaluate, for the first time, the utility of this geometric space in computing and visualizing the dependence structure of compound flooding drivers. To assess the effectiveness of this method as a hazard communication tool, we conducted a survey with a diverse group of end-users, including academic and non-academic respondents. The survey results provide insights into the perceptions of applicability of the Angles method and highlight its potential as an intuitive alternative to scatterplots in depicting the evolution of dependence in the non-stationary environment. This study emphasizes the importance of innovative visualization techniques in bridging the gap between scientific insights and practical applications, supporting more effective compound flood hazard communication.

 **1 Introduction**

Compound flooding from terrestrial (i.e. river discharge) and coastal (i.e. storm surge) drivers due to long-lasting (extra)tropical cyclones can have severe social and economic impacts for coastal communities around the globe (Zscheischler et al., 2018). Climate-driven increases in compound flooding present a growing sustainability challenge worldwide (Chan et al., 2024; Lai et al., 2021). A comprehensive hazard communication strategy is essential to engaging stakeholders and informing decision-making and mitigation efforts (Khan and Mishra, 2022), as well as supporting the UN Sustainable Development Goals (SDGs), specifically, SDG Target 11.5, which calls for reducing the adverse effects of natural disasters. Effective communication remains the main barrier to anticipating and responding to compound flood events (Kruczkiewicz et al., 2021). To date, many researchers have extensively explored the likelihood of co-occurrence of anomalously large river discharges and high sea water levels, at both local (Kim et al., 2023) and global (Couasnon et al., 2020) scales over the years (Radfar et al., 2024; Green et al., 2025). This type of analysis is well-grounded in the scientific literature and can be done by analysing the dependence structure of coinciding extremal samples of the variables of interest.

Bivariate analysis usually begins with a scatterplot that displays the variables of interest graphically in the variable space, where each axis represents a variable, and then calculate the correlation coefficient between them, e.g., the linear Pearsons' $r$ or the non-linear Kendall's $\tau$ or Spearman's $\rho$. This kind of visualization and computation of the dependence is prevalent in current scientific literature. To name a few examples, Robins et al. (Robins et al., 2021) plot coinciding extremes of river discharge and skew surge from two estuaries in the UK using a scatterplot and then calculate the Kendall's $\tau$, while Jane et al. (Jane et al., 2022) use the variable space for depicting the relationship between concurrent extremal values of storm surge and river discharge for three sites along the Texas Gulf Coast and subsequently compute the Kendall's $\tau$. Nasr et al. (Nasr et al., 2021) also follow Kendall's $\tau$ approach for quantifying dependence among different pairs of environmental extremes, including river discharge and storm surge across 36 coastal sites in the US. The variable space, however, does pose a limitation to studying the dependence structure, in the sense that it strongly places emphasis on the individual observations (subjects) themselves, which are denoted by points on the scatterplot, rather than the two variables for which inference is sought as generic entities. Yet better understanding of multivariate statistics and particularly of bivariate dependence calls for an effective and intuitive way of visualizing the relationship between variables with minimal focus on individual subjects. This is particularly important when the dimensionality of the problem increases, e.g., when an additional time dimension is introduced, to examine potential non-stationarities in the dependence structure of two variables. To overcome these limitations, we propose the Angles method, which uses Euclidean geometry to visualize the relationship between flood drivers in an intuitive way. Our approach aligns with established principles in visualization science that recognize different visualization methods serve distinct communication purposes (Munzner, 2014; Borgo et al., 2013). Current approaches for visualizing compound flood dependencies, including scatterplots and statistical measures, while mathematically sound, often struggle to effectively communicate evolving patterns to diverse end users. Copula-based approaches (Schoelzel and Friederichs, 2008) provide powerful statistical frameworks but can be mathematically complex

for non-specialists. The Angles method complements these approaches by offering a more intuitive visual representation specifically designed for communicating temporal evolution of dependencies. This perspective is especially important for compound flood hazard communication, where conveying evolving dependencies to non-technical audiences remains challenging. Although the present case study is bivariate, the same geometric logic scales to problems with three or more flood drivers by applying the subject-space projection to every driver pair in turn; this pairwise workflow is detailed later in Section 3.1.

In reality, multivariate statistics have a strong flavor of Euclidean geometry (Farnsworth, 2000; Friendly et al., 2013), which in turn can be an aid to unraveling the relationship between compound flooding drivers. Unlike the variable space and the scatterplot, if we think about the data in the "*subject space*" instead, where each subject (observation) of coinciding extremal pairs defines an axis (dimension), then the two variables can be represented as two points inside that space (Wickens, 2014). The idea of the subject space, although long known in statistical scientific literature, has rarely been explored in environmental sciences, let alone in studies concerning environmental extremes that may lead to flooding. In this note, we demonstrate how the use of this geometric space provides an alternative way of studying the dependence structure between environmental bivariate extremes, specifically river discharge and storm surge. What sets our work apart is its application in a multivariate non-stationary context, where it enhances hazard communication by providing an insightful means of visualizing evolving dependencies. Effective hazard communication is a critical component in disaster risk reduction (Fakhruddin et al., 2020; Pile et al., 2018) as it helps to inform, engage and educate vulnerable communities and stakeholders about the risks associated with natural hazards (Auermuller, 2019). This is of paramount importance to improve resilience against compound flooding, which is becoming an increasing threat to coastal communities in the changing climate (Bevacqua et al., 2020; Ghanbari et al., 2021). In this regard, the present study evaluates the effectiveness of the Angles method in visualizing evolving dependencies in compound flooding, emphasizing its potential for enhanced hazard communication. It should be emphasized that while statistical approaches like copulas provide sophisticated analytical frameworks for modeling compound flood hazards (Schoelzel and Friederichs, 2008), our focus is specifically on developing intuitive visualization techniques for effective hazard communication across diverse stakeholder groups. The Angles method is not meant to replace statistical methods like copula modeling, but rather to complement them by serving as an accessible first visual check of dependency relationships for broader audiences, including non-technical end users, before proceeding with more complex bivariate probability modeling.

## 2. Materials and Methods

### 2.1. Data Collection and Angles Method Development

For our analysis, we first used still water level data, composed of mean sea level, astronomical tide, and non-tidal residual, from tide gauges at Washington, DC, and Baltimore, MD, extracted from the GESLA3 database (Haigh et al., 2023). To extract the non-tidal residual, i.e., the storm surge, we performed tidal harmonic analysis on a rolling-year basis involving 60

major tidal constituents. Additionally, we utilized discharge data from rivers that drain to the respective tidal river outlet of each city, originating from the Global Runoff Data Center (GRDC) (Recknagel et al., 2023). Figure 1 illustrates the pairs of annual maximum discharge ($Q$) and respective maximum surge ($S$) within (+/-) a day of maximum-discharge timing (i.e., coinciding extremes) for the two coastal cities. In the case of the freshwater-influenced tige gauge at Washington, DC, the scatterplot is constructed from all 83 years of available measurements and the linear Pearson's $r$ correlation coefficient is

found to be 0.96 ($p$-value=0.000), while the non-linear Spearman's $\rho$ correlation coefficient is 0.84 ($p$-value=0.000). On the other end, 54 years of data from the Baltimore, MD gauge yield weaker correlations with Pearson's $r$ and Spearman's $\rho$ being 0.41 ($p$-value=0.005) and 0.52 ($p$-value=0.000), respectively. For both tide gauges, years with >20% missing data were excluded from both the scatterplots and the correlation analysis.

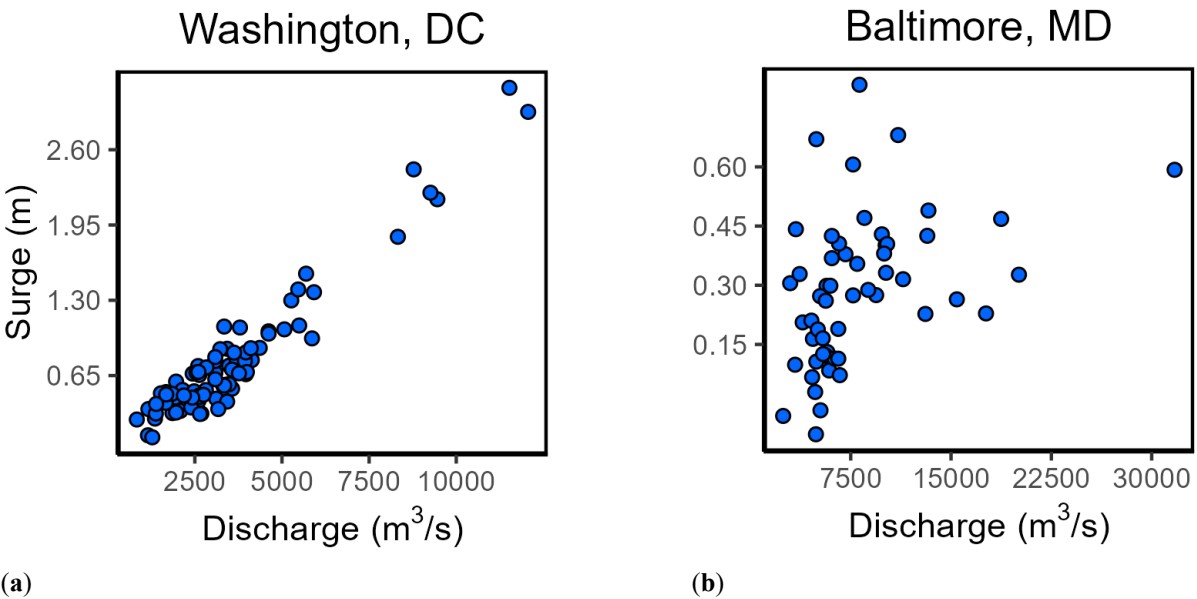

**(a)**                                                                                           **(b)**

**Figure 1. Scatterplots of discharge annual maxima and surge maxima within +/- 1 day of the maximum-discharge timing for: (a)**
**Washington, DC; (b) Baltimore, MD.**

In addition to traditional correlation analyses, the Angles method is applied to assess and visualize dependencies, offering a complementary perspective on compound flood dynamics. While the Angles method does not capture the full complexity of dependence structures (such as tail dependencies) that copula approaches can model, its primary strength lies in its visual intuitiveness for communicating evolving dependencies. Therefore, the method presented here is primarily

designed as a communication tool rather than a statistical modeling technique.

In the subject space of the data at Washington, DC, which consists of 83 axes (dimensions), equal to the number of pairs (subjects), discharge and storm surge can be defined by the two points:

$$Q = (4615, 3436, 3398, \ldots, 3086) \text{ and } S = (1.03, 0.88, 0.57, \ldots, 0.61) \tag{1}$$

Likewise, the data at Baltimore, MD, can be thought of as belonging to a 54-dimensional subject space where discharge and storm surge are simply two points:

$$Q = (2449, 5748, 2973, \ldots, 7673) \text{ and } S = (0.26, 0.20, -0.01, \ldots, 0.27) \tag{2}$$

Picturing variables $Q$ and $S$ in such high-dimensional spaces is obviously an impossible task for the human mind. Despite this limitation, the two points ($Q$ and $S$), together with the origin ($O$) of the subject space, form a 2-D plane which is easy to grasp, and thus discharge and storm surge can be plotted as two vectors (starting from the origin and extending to the respective point). For the sake of simplicity, we can center the two variables by subtracting the respective mean value of each variable so that the origin of the 2-D plane becomes zero, while their correlation and variances remain unchanged:

$$q = Q - \bar{Q} \text{ and } s = S - \bar{S} \tag{3}$$

From Euclidean geometry, we know that the length of a vector, e.g., the discharge vector ($\bar{q}$) is given by the following formula:

$$|\bar{q}| = \sqrt{q_1^2 + q_2^2 + q_3^2 + \ldots + q_N^2} \tag{4}$$

The squared length of $\bar{q}$ is then equivalent to the sum of squared deviations from the zero mean:

$$|\bar{q}|^2 = \sum_{i=1}^{N} q_i^2 \tag{5}$$

Hence, the length of vector $\bar{q}$ is directly related to the unbiased estimator of the standard deviation of discharge population:

$$\sigma_q = \frac{|\bar{q}|}{\sqrt{N-1}} \tag{6}$$

Correspondingly, it follows the same for the standard deviation of surge population:

$$\sigma_s = \frac{|\bar{s}|}{\sqrt{N-1}} \tag{7}$$

Euclidean geometry, and particularly trigonometry, indicates also that the cosine of the angle between two vectors is equal to their dot product ($\cdot$) over the product of their lengths, as shown below:

$$\cos(\theta) = \frac{\bar{q} \cdot \bar{s}}{|\bar{q}||\bar{s}|} \tag{8}$$

where $\theta = \angle(\bar{q}, \bar{s})$, and $\bar{q} \cdot \bar{s} = q_1 s_1 + q_2 s_2 + q_3 s_3 + \dots q_N s_N$. It is now easy to see that the expression in Equation 8 matches that of Pearson's $r$ correlation coefficient:

$$r = \cos(\theta) = \frac{\bar{q} \cdot \bar{s}}{|\bar{q}||\bar{s}|} = \frac{\sum_{i=1}^{N} q_i s_i}{\sqrt{\left(\sum_{i=1}^{N} q_i^2\right)\sum_{i=1}^{N} s_i^2}} \tag{9}$$

In the subject space, uncorrelated discharge and surge variables are displayed as perpendicular vectors ($\theta = 90°$), whereas correlated discharge and surge variables are displayed as collinear vectors ($\theta = 0°$ or $\theta = 180°$). Many times, in multivariate statistics, variables are not only centered around zero but also scaled by dividing them with their standard deviation. In such an instance, the standard deviation of each variable becomes then one and thus from Equations 6 and 7 it follows that the vectors $\bar{q}$ and $\bar{s}$ have the same length, only dependent on N. For convenience, one may choose to work with

vectors of unit length $|\bar{q}| = |\bar{s}| = 1$ and hence the constant $\sqrt{N-1}$ can be neglected – then, the only characteristic of the two vectors that truly matters is the angle between them. Consequently, a greater angle $\theta$, i.e., a smaller Pearson's $r$ will lead to a bigger parallelogram area between the two vectors since Euclidean geometry suggests that:

$$\text{Area}_{\text{parallelogram}} = |\bar{q}||\bar{s}|\sin(\theta) = \sin(\theta) = \sqrt{1 - (\cos[\theta])^2} = \sqrt{1 - r^2} \tag{10}$$

While we present the geometric interpretation using Pearson's correlation coefficient in this section, it is important to acknowledge its limitations, including problems of existence in certain cases, restriction to linear associations between

variables, and lack of invariance under monotonic transformations (Salvadori et al., 2007; De Michele et al., 2005; Serinaldi et al., 2022). To address these limitations, this approach can be extended to Spearman's rank correlation coefficient, which offers advantages in handling non-linear relationships, maintains invariance under monotonic transformations, and provides more robust estimations when dealing with outliers or non-normal distributions. The complete derivation of the geometric interpretation using Spearman's correlation is presented in Appendix A.

**2.2. Survey Design and Implementation**

To assess the end users' perceptions of the effectiveness of the Angles method and the subject space for visualizing coastal compound flooding (CCF) dependencies, we conducted a survey targeting a diverse group of end users. We conducted an online survey between July 10 and September 10, 2024 and distributed it through multiple channels to reach a broad audience. These channels included email lists, members of the working Group 4 of the Cooperative Institute for Research to

Operations in Hydrology (CIROH) institution, which focuses on impact-based decision-making research, stakeholders from the NOAA project "Coastal Nature-Based Solutions to Mitigate Flood Impacts and Enhance Resilience," and the network of

the Program for Local Adaptation to Climate Effects (PLACE). This distribution strategy allowed us to gather input from a wide range of respondents, including members from the academia, industry, non-governmental organizations (NGOs), and state, federal, and local government agencies. Our evaluation framework follows established principles in visualization science (Munzner, 2014) that recognize the importance of assessing visualization techniques based on their intended purpose and audience. We designed a sequential comparative evaluation to assess both immediate understanding and effectiveness for communicating specific concepts like non-stationarity. This approach allows for a comprehensive assessment of the method's strengths and limitations across different stakeholder groups.

The questionnaire gauged the respondents' familiarity with CCF dependencies, the clarity of non-stationarity concepts, and the effectiveness of the Angles method in communicating compound hazard. Likert scale questions were used to capture the degree of agreement or disagreement on various aspects of the Angles method, including its understandability, applicability, and perceived usefulness in CCF hazard communication. The responses are subsequently grouped into two categories: academic and non-academic respondents. This classification is used to evaluate the differing perceptions of the proposed Angles method between these two groups. Academic respondents primarily included researchers, faculty, and students from various universities, while non-academic respondents comprised professionals from the industry, government agencies, and NGOs. This segmentation clarifies how familiarity, relevance, and clarity vary between the two sectors.

## 3. Results and discussion

### 3.1. Application of the Angles method for visualizing CCF dependencies

Figure 2 shows the two variables ($Q$ and $S$) represented as unit-length vectors ($\bar{q}$ and $\bar{s}$) on a 2-D plane of the subject space with zero origin. As opposed to scatterplots, these graphs show the two variables as general entities rather than individual observations. The angle between the two vectors is proportional to how dependent they are, as shown in Equations 9 and 10. Note the small angle in Washington, DC and the large angle in Baltimore, MD, which correspond to the respective parallelogram areas. In fact, for the two cases, inserting the vector lengths and the dot product of the discharge and surge vectors into Equation 8 yields $\cos(\theta) = r = 0.96 \Rightarrow \theta \sim 16°$ and $\cos(\theta) = r = 0.41 \Rightarrow \theta \sim 66°$, respectively. In Washington, DC, the smaller angle between the unit vectors indicates a tighter interplay between river discharge and storm surge. This suggests a higher correlation and potential for severe compound flooding events. Conversely, in Baltimore, MD, the larger angle suggests a lower degree of correlation. Thus, although both drivers matter, they coincide less often to generate severe compound events.

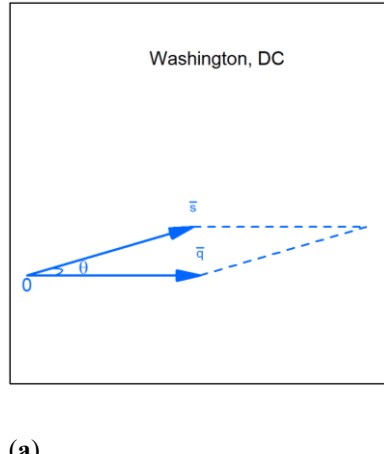

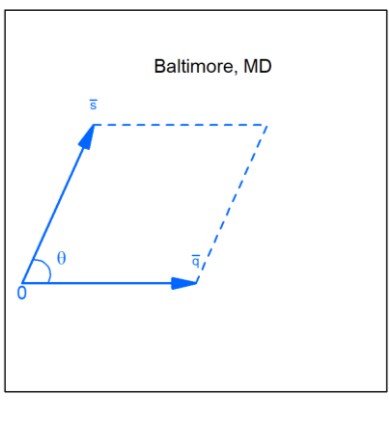

**(a)**                                                                                          **(b)**

**Figure 2. Discharge (Q) and surge (S) variables represented as unit-length vectors in the subject space for: (a) Washington, DC where cos[$\theta$] = 0.96; (b) Baltimore, MD with cos[$\theta$] = 0.41.**

The subject space provides an effective approach when dealing with more than two variables, e.g., multi-driver compound flooding from discharge, surge, precipitation, and wind waves. It is inherently difficult to illustrate 4-dimensional scatterplots, and the interactions of multiple flooding drivers cannot be visually captured by such a plot. In such cases, Euclidean geometry offers a systematic solution through pairwise analysis. Each pair of flood drivers can be represented as vectors in a 2-D plane, with their angular separation revealing their dependence structure. This pairwise projection approach allows for clear visualization and interpretation of relationships between multiple flood drivers, overcoming the limitations of multi-dimensional scatterplots while maintaining geometric intuition.

In addition, plots such as those in Figure 2 can be a complementary tool to visualize changes in the dependence structure over time. If human-induced climate change is making the co-occurrence of flood drivers more likely (Wahl et al., 2015), this can be visualized by a frame with a shrinking angle $\theta$. For example, Figure 3 illustrates a scatterplot of bivariate sampling where the *y* axis shows annual maxima still water levels at Galveston Pier 21, TX, while the *x* axis represents co-occurring (+/- 5 days) maxima of discharge at Buffalo Bayou which drains into Galveston Bay – data from different time periods are highlighted with different colors. From Figure 3 alone, it is not evident if the dependence between the two flooding drivers is getting stronger with time. Specifically, it appears rather hard to determine (by a mere visualization) whether the correlation coefficient from 1972-1996 is greater than that of the period from 1997-2022. In many times, the scatterplot fails to reveal evolving patterns of dependence.

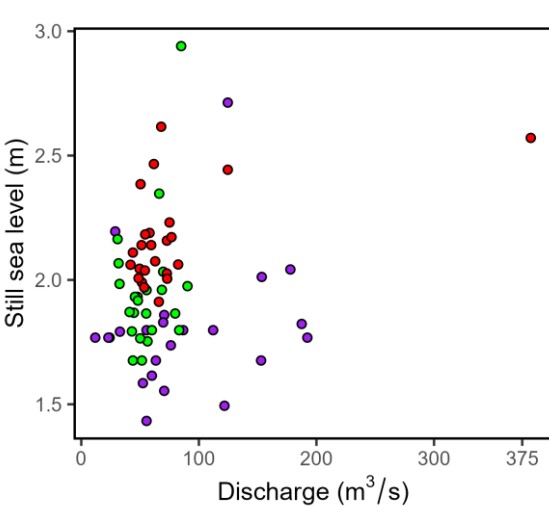

**Figure 3. Scatterplot of annual maxima sea levels and discharge maxima within +/- 5 days of the maximum-sea-level timing for Galveston Bay, TX. Pairs are colored based on period of observation.**

200      On the contrary, visualizing the variables as unit-length centred vectors again, where the pair-wise angle is the only key characteristic between them, allows us to infer that the dependence between extreme sea levels and river discharge at Galveston Bay has been increasing over time (observe the shrinking angle $\theta$ in Figure 4). From Figure 4, we observe that the correlation coefficient of the period 1997-2022 ($r = 0.42$, $p$-value = 0.035) is greater than that of 1972-1996 $r = 0.21$, $p$-value = 0.393), which is reflected in the smaller angle $\theta$ (thus, the larger cosine) in the more recent period. This evolving trend is a

205   sign of non-stationarity in dependence structure, which is per se a difficult concept to communicate to a variety of stakeholder groups. Presenting this simple graph underscores the importance of considering temporal changes in dependence structure when planning and implementing flood risk management strategies. This dynamic understanding aligns with adaptive management principles in coastal engineering. It necessitates continuous monitoring and re-assessment of flood risks considering potential non-stationarity in hydrodynamic, hydrological and meteorological relationships.

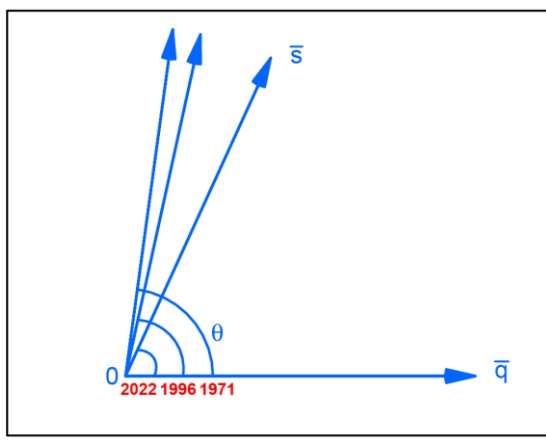

**Figure 4. Subject space showing a stronger dependence between sea levels and discharge over non-overlapping time periods (end year shown in red), i.e., a shrinking angle $\theta$ between the two vectors at Galveston Bay, TX.**

Another important aspect within compound flooding framework is non-stationarities of the dependence structure among flood drivers. In a warming climate, assuming stationarity can mask climate-driven shifts in variability (Milly et al., 2008), as it does not consider increasing changes in variation of flood drivers due to climate change (Kim et al., 2018). Natural climatic variability and anthropogenic change both introduce non-stationarity (Galiatsatou and Prinos, 2011), and ignoring it can mislead multivariate assessments (Radfar and Galiatsatou, 2023; Corbella and Stretch, 2012). Non-stationarity also influences the dependence structures among compound flood drivers over time (Naseri and Hummel, 2022); dynamic, non-stationary copulas offer one solution (Pirani and Najafi, 2023), yet their complexity means most studies default to moving-window analyses or a stationary assumption (Radfar et al., 2023).

Public perception of this impact is even more challenging. The expected annual economic losses due to compound flooding damage amount to billions of dollars. Yet, the knowledge about non-stationarity in compound flood drivers is still very limited among practitioners and stakeholders and this could hinder proper preparedness and mitigation efforts against this increasing risk to coastal communities. To disseminate information about changing dependence structures to the target audience, it would be necessary to adopt effective communication approaches. Figure 5 illustrates how non-stationarity in the dependence of the two variables over multiple, possibly overlapping time periods, can be effectively visualized with the use of the subject space: the angle $\theta$ shrinks from obtuse in 1950-1991 (past) to acute in 1982-2021 (present), indicating that the negative correlation between discharge and sea level extremes has gradually evolved into a strong positive dependence over time. A unique characteristic of semi-circular representation of Figure 5 is its capability to encompass equal, unit-length vectors to clearly depict an evolving correlation among flood drivers over time. This easy-to-follow visualization technique could help overcome challenges in communicating with non-experts, aiding in their better understanding of the shifting dependence between multiple flood drivers, and ultimately, motivating them about compound flood mitigation efforts. It is expected that such simple visualization efforts will better reflect climate change effects and emphasize the need for resilient

infrastructure and adaptive measures to safeguard against floods. Ultimately, this enables vulnerable coastal communities to remain resilient and sustainable in the face of a warming climate, which is an overarching objective of SDGs 11 (Sustainable Cities and Communities) and 13 (Climate Action).

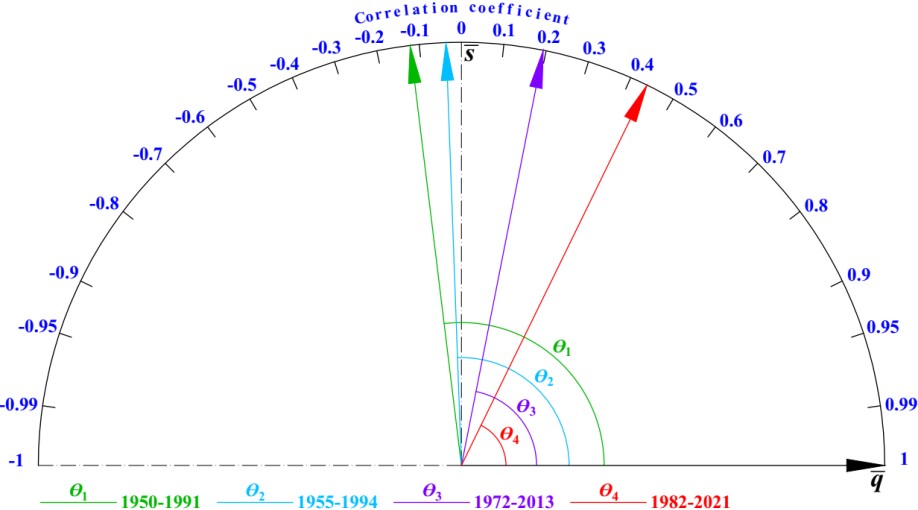

**Figure 5. Subject space showing a stronger dependence between sea levels and discharge across multiple overlapping time periods, i.e., a shrinking angle θ between the two vectors at Galveston Bay, TX. Observe how an obtuse angle, i.e., a negative correlation in the past, gradually transforms into an acute angle indicating strong positive correlation.**

### 3.2. Survey Results and Analysis

The survey collected 91 complete responses. The top panel in Figure 6 shows a world map highlighting the global reach of the survey, with participants spread across multiple continents. Most respondents were from the United States (n = 64). Other contributions came from the United Kingdom (5), India (4), France (3), the Netherlands, Spain, and Australia (2 each), plus nine additional countries with one respondent apiece. This distribution shows global engagement, and captures perspectives from multiple regions and sectors. In the United States, the survey responses came from 20 states, with the highest numbers reported in Mississippi (14), Alabama (13), and Florida (10) along the Gulf Coast. This concentration is primarily due to the survey distribution channels, which are closely connected to organizations and projects in this region.

The responses are subsequently grouped into two categories: academic and non-academic respondents. This classification is used to evaluate the differing perceptions of the proposed Angles method between these two groups. Academic respondents primarily included researchers, faculty, and students from various universities, while non-academic respondents comprised professionals from the industry, government agencies, and NGOs. This segmentation allows us to explore how familiarity, relevance, and clarity of the Angles method differed across these distinct sectors. The bottom panel in Figure 6 presents bar charts comparing the responses of academic (*n* = 44) and non-academic (*n* = 47) respondents to

questions regarding their familiarity with CCF, the relevance of CCF to their work, years of experience, and familiarity with the concept of non-stationarity. When "very well" and "extremely well" are combined, 68.1 % of non-academics versus 59 % of academics report high familiarity. The relevance of CCF to respondents' work was high for both groups, with 56.8% of academics and 70.2% of non-academics reporting it as "extremely" relevant. Similarly, regarding years of experience, non-academics showed a higher proportion (59.6%) with extensive experience compared to academics (47.7%). Interestingly, familiarity with non-stationarity concepts revealed a more pronounced divide, with 25% of academics reporting being "extremely well" familiar with non-stationarity, compared to only 14.9% of non-academics in the same category. This difference becomes even more pronounced when considering those who are less familiar with the concept. Notably, 34% of non-academics reported being "not well at all" familiar with non-stationarity, which is significantly higher than the 13.6% of academics in the same category. This disparity might reflect the theoretical and complex nature of non-stationarity, which may be more frequently encountered in academic research.

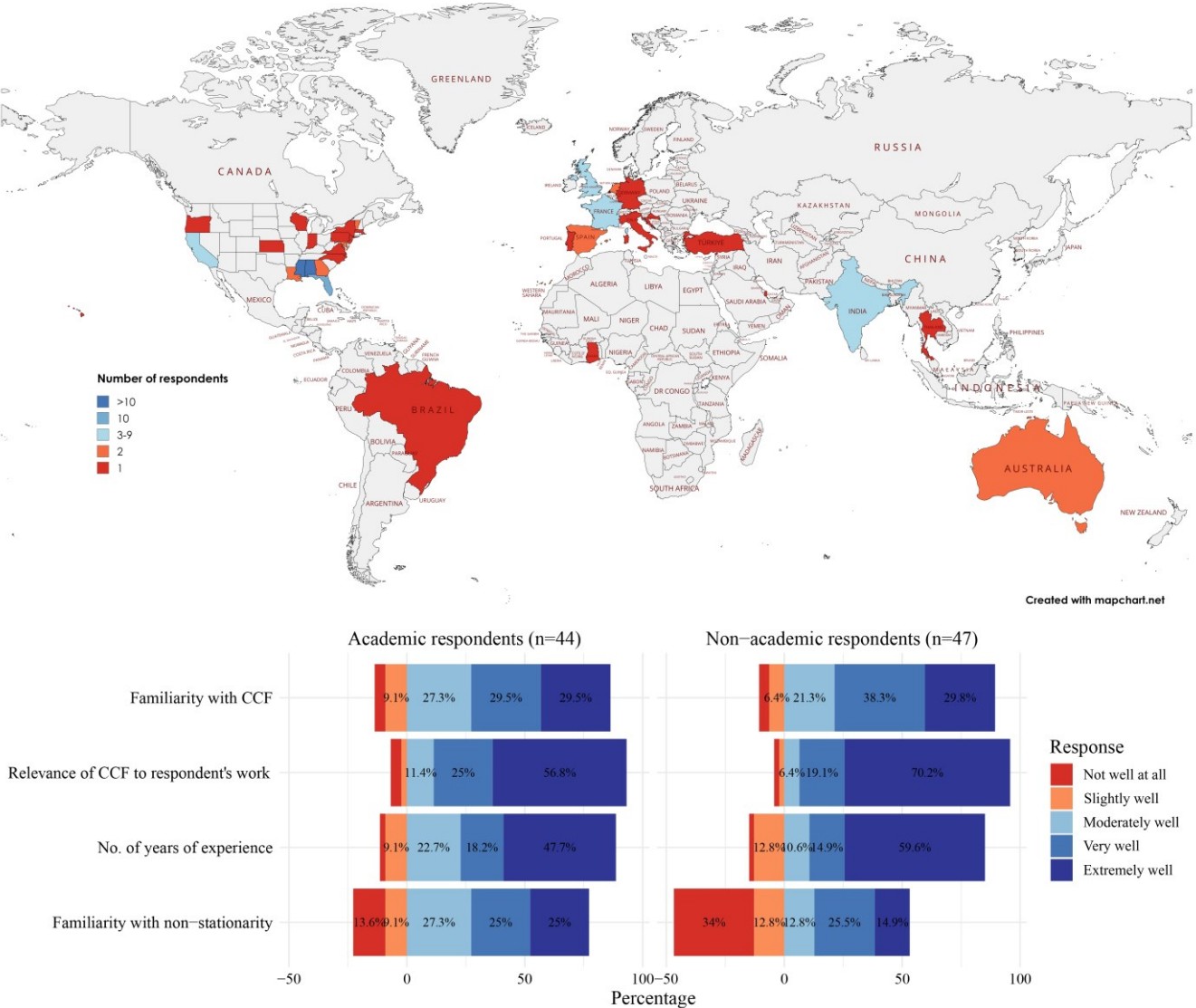

**Figure 6. Geographic distribution and knowledge assessment of the survey respondents. Top panel: World map showing the geographic distribution of survey respondents (91 total respondents) who participated in the study on compound flood risk communication. Countries are color-coded based on the number of respondents, ranging from 1 to over 10. Bottom panel: Bar charts depicting the Likert scale responses from academic (44 respondents) and non-academic (47 respondents) respondents on their familiarity with coastal compound flooding (CCF), relevance of CCF to their work, years of experience, and familiarity with non-stationarity concept.**

Figure 7 presents a detailed comparison of academic and non-academic respondents' perceptions of various aspects of CCF hazard communication, utilizing Likert scale responses. The bar chart highlights how these groups responded to various aspects of the Angles method, including correlation understandability, non-stationarity clarity, effectiveness in CCF hazard communication, and the likelihood of applying the method in their work or public communication.

First, we evaluated the understandability of correlation using strictly numerical values (i.e., correlation coefficients) versus the Angles method, which incorporates numerical values into a visual representation. For the numerical approach, academic respondents showed a higher level of agreement (50% agree or strongly agree) compared to non-academics (35.9%). When assessing the Angles method, academic respondents maintained a similar level of agreement, with 68.2% at least slightly agree with its understandability. However, among non-academics, the level of high agreement ("agree" and "strongly agree") dropped to 19.2%. This shift can partly be related to the findings from Figure 6, where most respondents reported significant familiarity with compound flooding, and accordingly, the concept of dependence between flood drivers. This familiarity suggests that respondents are accustomed to traditional correlation coefficients, which may bias them toward these conventional methods rather than accepting a complementary visual approach for communicating a rather simple concept of correlation between flood drivers.

While comparing a method like the Angles approach, that incorporates familiar numerical values into a visual representation, against strictly numerical values may seem unconventional, especially among experts, it is crucial for establishing a benchmark of the new method's capabilities. Previous studies have consistently demonstrated that when communicating with the public, numbers, graphs, and technical concepts often fall short in effectively conveying the importance of hazards and risks (Morrow et al., 2015; Kuser Olsen et al., 2018). Visualization has proven to be a key tool for enhancing understanding, engagement, and decision-making (Atasoy et al., 2022; Colle et al., 2023). Thus, evaluating the Angles method against traditional numerical values was necessary to understand how well it performs relative to established approaches, even within an expert perspective. The results, although showing lower levels of familiarity among non-academics, remain promising and acceptable, underscoring the Angles method's potential as a more intuitive alternative that could bridge gaps in understanding when deployed beyond expert audiences.

Next, building upon the initial comparison, and to ensure a one-to-one comparison, we evaluated the Angles method against scatterplots for representing non-stationarity (here, in the form of evolving dependencies). For scatterplots, 40.9% of academics believe the scatterplot clearly depicts variations in dependence (agreeing or strongly agreeing), whereas only 27.7% of non-academics indicated such. However, when considering clarity for a potential audience, both groups expressed lower confidence, with only 11.3% of academics and 4.3% of non-academics agreeing or strongly agreeing. These findings clearly imply the complexity of the non-stationarity concept and the challenge of communicating it with non-academics and the audience of both groups. Interestingly, both the academics and non-academics reported that the Angles method offered improved clarity for them and their potential audiences. For respondent clarity, 77.3% of academics agreed at least slightly, compared to 68.1% of non-academics. For presumed clarity to a hypothetical prospective audience, the Angles method enhanced the level of agreeing or strongly agreeing from 11.3% to 34.1% for academics, and from 4.3% to 23.4% for non-academics, compared to scatterplots.

Finally, Figure 7 further illustrates the effectiveness of the Angles method in CCF hazard communication and the likelihood of applying it in professional settings. Regarding the effectiveness in CCF hazard communication, academic respondents appeared more positive, with 31.8% agreeing or strongly agreeing, versus 23.4% of non-academics. Academic

respondents showed a strong consensus on the method's practical application, with 75% expressing a likelihood of applying it in their work or research. In contrast, non-academic respondents were more divided, with 53.2% expressing some likelihood of applying it, but with a notable 21.8% disagreeing or remaining neutral, suggesting a hesitancy to adopt the method without further familiarization. For public communication, both groups turned into higher strongly agreement and lower agreement.

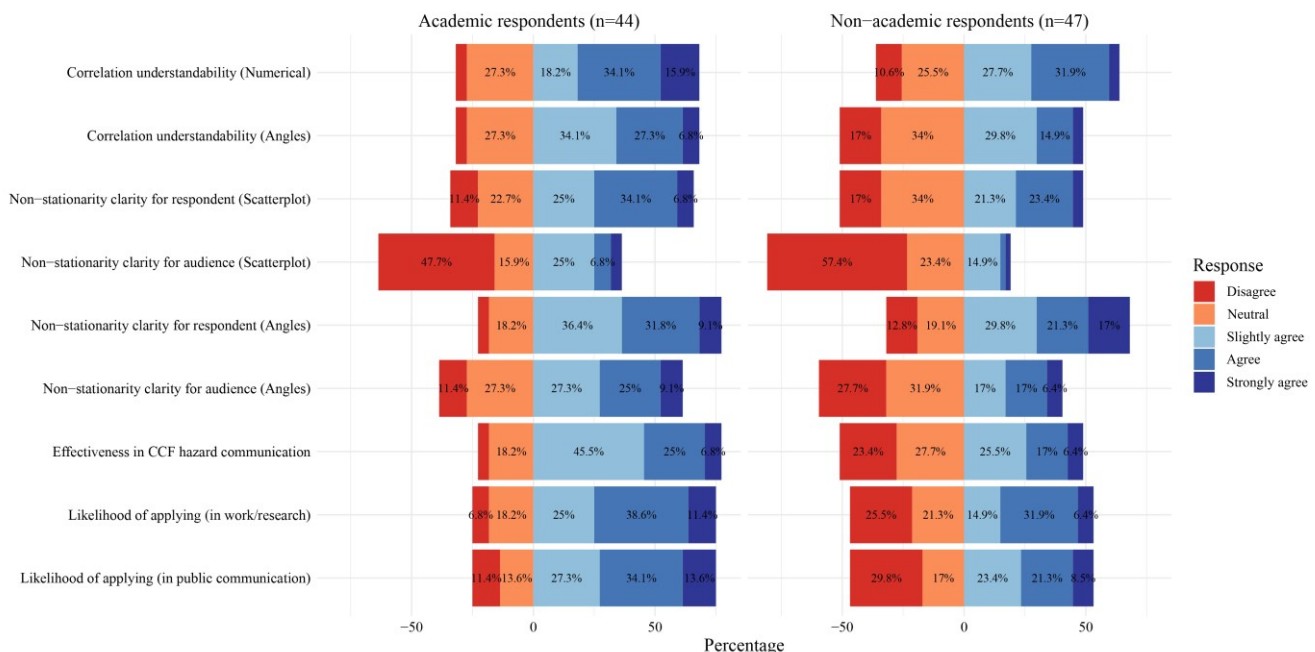

**Figure 7. Bar chart of Likert scale responses comparing academic (*n* = 44) and non-academic (*n* = 47) perceptions of coastal compound flooding (CCF) hazard communication. The figure shows the percentage distribution of responses on correlation understandability, non-stationarity clarity, effectiveness in hazard communication, and likelihood of applying in work/research and public communication.**

Figure 8 illustrates the relationships between various aspects of CCF understanding, hazard communication, and application likelihood. For academic respondents, the figure shows that those with greater familiarity with CCF and those who find CCF highly relevant to their work tend to believe that scatterplots are not effective tools for communicating non-stationarity to audiences, as indicated by the negative correlations. This pattern is similarly observed among respondents with higher degrees, more experience, and familiarity with non-stationarity concepts, suggesting a general skepticism toward traditional scatterplot use in conveying complex, evolving relationships.

Conversely, when the Angles method is used to represent non-stationarity, there is a notable positive shift in correlations. This significant positive relationship suggests that academic respondents who were initially critical of scatterplots found the Angles method to be a more effective visual tool for communicating non-stationarity. This shift underscores the potential of the Angles approach to address perceived gaps in traditional hazard communication methods

among those with more advanced knowledge and expertise, and highlights its value in enhancing understanding of the evolution of the dependency of flood hazard drivers.

Among non-academic respondents, varying correlations are observed. This divergence might reflect differences in how
335 these factors influence openness to the CCF communication methods in academic versus practical settings. Notably, the degree of non-academic respondents shows moderate positive correlations with years of experience in hydrologic or hydrodynamic fields (0.45) and familiarity with CCF (0.31), but weak or negative correlations with most other factors. This could suggest that while higher degrees are associated with more experience and familiarity, they do not necessarily translate to increased clarity or the likelihood of applying new communication methods. The figure reveals that for non-academic
340 respondents, the relevance of CCF to their work shows positive correlations with most factors. It exhibits stronger positive correlations with the Angles method compared to traditional methods like numerical values or scatterplots. This suggests that non-academics who find CCF relevant to their work are more likely to perceive the Angles method as an effective tool for understanding and communicating complex dependencies, compared to more traditional approaches.

It is important to note that using language like "new," "groundbreaking," or "different" can sometimes bias people
345 against trying or adopting unfamiliar methods, as they tend to prefer what is familiar. In future discussions with audiences that may be hesitant to adopt the Angles method, emphasizing that it builds upon familiar concepts like correlation coefficients by adding a visual element, rather than contrasting with them, may increase the likelihood of its adoption.

Another pattern observed in the figure is that respondents who find each of the methods clear for themselves also believe it would be clear for their audience. This relationship is particularly pronounced among non-academic respondents,
350 where there are significantly stronger positive correlations between the clarity of the methods for the respondent and its perceived clarity for the audience. This suggests that non-academics who understand these methods well are more confident in their effectiveness as a communication tool for broader audiences.

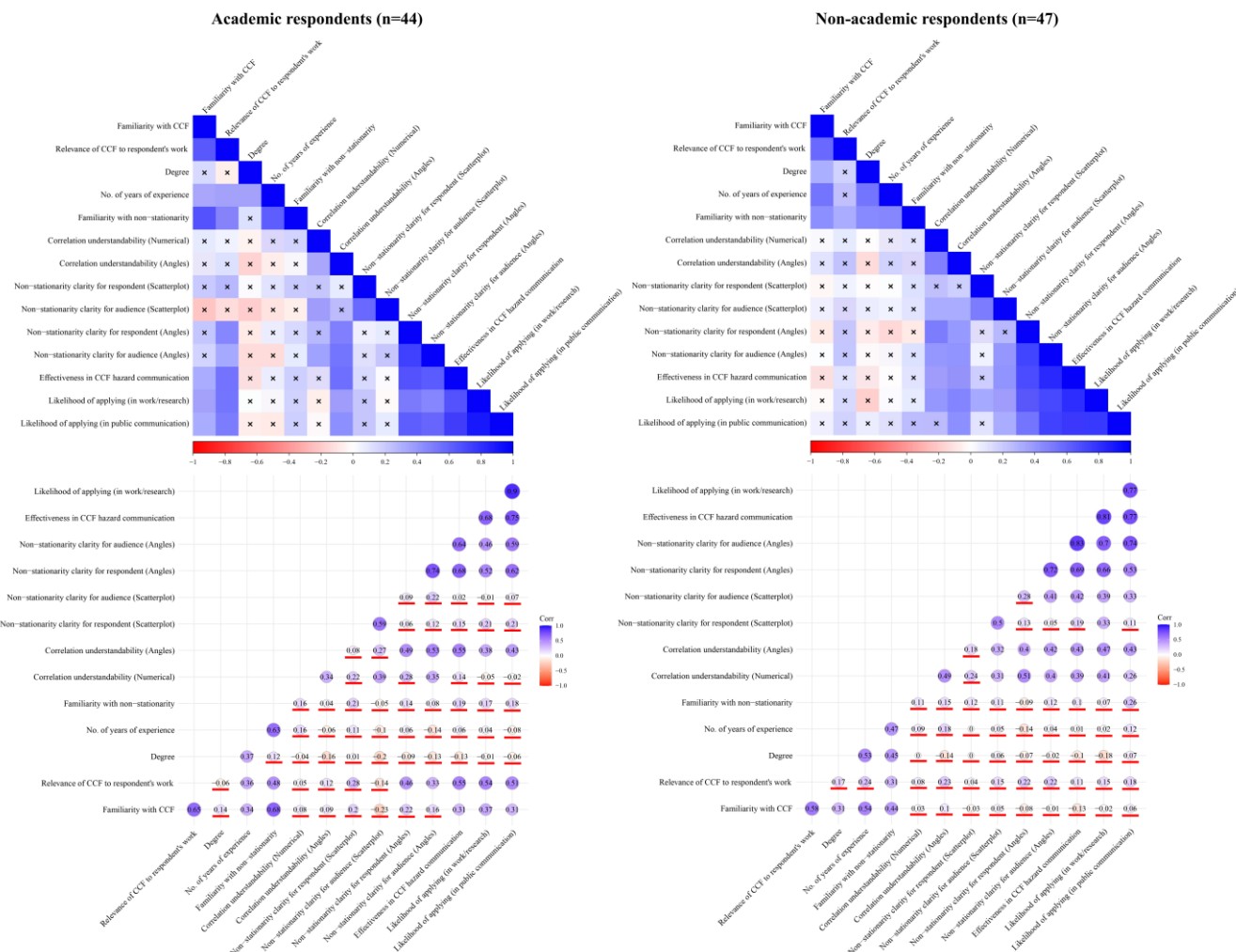

**Figure 8.** Correlation matrices comparing academic (*n* = 44) and non-academic respondents (*n* = 47) on familiarity, understanding, effectiveness, and likelihood of practical application of the Angles method compared to the traditional approaches (correlation values and scatterplots) for CCF hazard communication. The heatmaps display correlation coefficients, with color gradients indicating the strength and direction of correlations (blue for positive, red for negative). Circle sizes in the lower triangle represent correlation magnitudes. Correlations with *p*-values>0.05 are marked with crosses (top panel) and underlines (bottom panel) to indicate statistical insignificance.

## 4. Conclusions

This study evaluates the Angles method as a complementary visual tool for communicating evolving dependencies in CCF hazards. Rather than replacing sophisticated statistical methods like copulas, the Angles method serves a distinct purpose in making temporal patterns of dependence more accessible to diverse stakeholder groups, particularly those without technical backgrounds. The Angles method leverages Euclidean geometry to transform numerical dependencies into visual angles, where each angle represents the relationship between flood drivers. This geometric representation allows for a more intuitive

understanding of the complex dependencies compared to traditional numerical correlations (Section 2). By augmenting statistical relationships with visual patterns, the Angles method provides an accessible way to identify changes in dependencies over time, making it a powerful tool for non-stationary hazard communication (Section 3.1). Our findings reveal that the Angles method offers sensible advantages over traditional scatterplots, especially in enhancing the understanding and communication of evolving dependencies among CCF drivers (Figure 7).

The survey results demonstrated that the method was primarily evaluated among a group of experienced respondents from the academic, industry, and government sectors (Figure 6). Academic respondents generally reported higher familiarity with CCF dependencies and perceived the Angles method as more effective in enhancing communication of dependencies between compound flood drivers compared to traditional approaches (Figure 7). In contrast, non-academic respondents exhibited varying levels of familiarity and clarity, indicating a need for tailored communication strategies when presenting new methods like the Angles approach to diverse stakeholder groups.

Comparisons between the Angles method and scatterplots revealed that the Angles method provided a clearer and more intuitive representation of non-stationarity, particularly for academic respondents (Figure 7). This suggests that the Angles method can effectively fill existing gaps in traditional hazard communication by offering a visual alternative that captures the dynamic nature of CCF dependencies. Non-academic respondents also showed more positive correlations between the relevance of the Angles method to their work compared to traditional methods (Figure 8), indicating its potential alignment with practical needs in real-world flood management contexts.

The survey also highlighted a pattern where those who found the Angles method clear for themselves believed it would also be clear for others, with this effect being particularly pronounced among non-academic respondents. This underscores the Angles method's potential to facilitate effective communication beyond expert audiences, bridging gaps between scientific insights and practical applications in flood hazard communication.

These findings highlight the opportunity to further develop the Angles method for communication with a non-technical audience. Given that the current evaluation focused on experienced respondents (Figure 6), future studies should explore the effectiveness of the Angles method with broader audiences, including the public and students. Engaging educational initiatives, such as those supported by the Scientific Research and Education Network (SciREN; https://sciren.ua.edu/), would provide valuable insights into how well this method communicates complex flood hazard information to non-expert audiences. Such evaluations would not only validate the Angles method's utility across different groups but also enhance its role in scientific education and public understanding of environmental hazards.

**Appendix A: Geometric Interpretation Using Spearman's Rank Correlation**

The geometric interpretation presented in Section 2 can be extended to Spearman's rank correlation coefficient ($\rho$), which offers several advantages over Pearson's correlation ($r$), including better handling of non-linear relationships and invariance under monotonic transformations. Here we present the complete derivation:

Instead of working with the original variables directly, we first transform the data into ranks and then into pseudo-observations:

$$q^S = F_Q(Q) = \frac{\text{rank}(Q)}{N+1}, s^S = F_S(S) = \frac{\text{rank}(S)}{N+1} \tag{A1}$$

where $q^S$ and $s^S$ are the pseudo-observations representing the probabilistic ranks of discharge and surge respectively, $F_Q$ and $F_S$ are the empirical cumulative distribution functions, $\text{rank}(Q)$ and $\text{rank}(S)$ are the ranks of observations, and $N$ is the sample size.

Similar to the Pearson-based approach, we can represent these transformed variables as vectors in the subject space. The length of these vectors can be calculated as:

$$\left|\overline{q^s}\right| = \sqrt{(q_1^S)^2 + (q_2^S)^2 + (q_3^S)^2 + \cdots + (q_N^S)^2} \tag{A2}$$

with the squared length being:

$$\left|\overline{q^s}\right|^2 = \sum_{i=1}^{N} (q_i^S)^2 \tag{A3}$$

The standard deviation of the transformed variables is given by:

$$\sigma_{q^S} = \frac{\left|\overline{q^s}\right|}{\sqrt{N-1}}, \sigma_{s^S} = \frac{\left|\overline{s^s}\right|}{\sqrt{N-1}} \tag{A4}$$

The Spearman correlation coefficient ($\rho$) can then be expressed geometrically as the cosine of the angle between the transformed vectors:

$$\rho = \cos(\theta^S) = \frac{\sum_{i=1}^{N} q_i^S s_i^S}{\sqrt{\sum_{i=1}^{N}(q_i^S)^2 \sum_{i=1}^{N}(s_i^S)^2}} \tag{A5}$$

This formulation maintains all the geometric properties discussed in Section 2, including the relationship between the angle $\theta$ and the correlation coefficient, but offers additional robustness to non-linear relationships between the original variables $Q$ and $S$. Like the Pearson-based approach, uncorrelated variables are represented by perpendicular vectors ($\theta = 90°$), while perfectly correlated variables have parallel vectors ($\theta = 0°$ or $180°$).

The key advantage of this Spearman-based geometric interpretation is that it captures monotonic relationships between the variables, not just linear ones, making it particularly suitable for analyzing compound flooding drivers that may exhibit complex, non-linear dependencies. Additionally, the rank transformation makes the approach less sensitive to outliers and more appropriate for non-normally distributed data, which is often encountered in environmental extremes.

**Code availability**

The code used for the analysis and visualization in this study, including the implementation of the Angles method for visualizing compound flood drivers and the analysis of survey results, is available in the GitHub repository: https://github.com/sradfar/CFnonStatViz.

**Data availability**

The data supporting the findings of this study, including the survey results and relevant datasets used for visualizing the dependence structure of compound flood drivers, are available in the GitHub repository. Additional data, such as river discharge and sea level data, can be accessed from publicly available sources like the Global Runoff Data Center (GRDC) and the GESLA3 database. The GitHub repository can be accessed at: https://github.com/sradfar/CFnonStatViz.

**Author contribution**

GB, SR, and HMR conceptualized the study. SR and GB contributed to the methodology, with GB and SR focusing on the statistical model and SR handling the survey analysis. WS, LL, and ARN assisted with preparing, reviewing, and distributing the survey. Data curation and analysis were conducted by SR and GB. SR and GB wrote the original draft, and all co-authors reviewed and provided feedback on the manuscript.

**Competing interests**

The authors declare that they have no conflict of interest.

**Ethical statement**

This research received ethical approval from The University of Alabama's Institutional Review Board (IRB). Informed consent was obtained from all participants prior to participation, and the survey was conducted anonymously to ensure confidentiality.

**Acknowledgment**

We would like to thank all the individuals who participated in our survey for their valuable input.

**Financial support**

This study is funded by National Science Foundation awards #2223893 and #2238000. Partial support is also provided by National Oceanic and Atmospheric Administration (NOAA), awarded to the Cooperative Institute for Research on
Hydrology (CIROH) through the NOAA Cooperative Agreement with The University of Alabama, NA22NWS4320003.

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
