# Peer review of "The value of visualization in improving compound flood hazard communication: A complementary perspective through a Euclidean Geometry lens"

_Geoscience Communication, 2024_

## Author Comment (AC1)

General comments

The manuscript introduces the Angles method (based on Euclidean geometry of the so-called "subject space") for visualizing the dependence structure of compound flooding drivers. Then it is evaluated the utility of the methodology for risk communication through a survey with diverse group of end-users, including academic and non-academic respondents.

*Answer*: We sincerely appreciate your time and constructive feedback on our manuscript. We have carefully addressed your comments and made the necessary updates to the manuscript. We hope the revisions align with your suggestions and strengthen the clarity and impact of our work. Thank you for your valuable input.

**C#1)** The Authors use a geometrical interpretation of Pearson's correlation coefficient (Eq.s 4-9). This issue is interesting and promising.

However, the Pearson's correlation coefficient has some weaknesses: 1) problems of existence [see e.g., Salvadori er al. 2007 and De Michele et al. 2005]; 2) represent the linear association between the variables (as highlighted also by the Authors); 3) It is not invariant under monotonous transformation (only linear ones), issue of great importance for the application of Sklar's theorem and thus copulas applications (see Salvadori er al. 2007). In this respect, why not using the Spearman correlation coefficient? According to the connection between the Pearson's correlation coefficient and the Spearman's one, you can write easily Eq.s 3-9 in terms of the pseudo-observations / transformed variables F(Q) and F(S). I suggest to develop this case in substitution (better) or alternative.

*Answer*: Thank you for this insightful comment about the limitations of Pearson's correlation and the suggestion to use Spearman's correlation coefficient. We agree that Pearson's correlation has several important limitations as you've noted, particularly regarding problems of existence, and its restriction to linear associations and lack of invariance under monotonic transformations. To address these limitations, we have made two major updates to the manuscript:

1) At the end of Section 2, we now include a paragraph acknowledging these limitations and pointing readers to an alternative geometric interpretation using Spearman's rank correlation, which addresses many of these concerns:

> "*While we present the geometric interpretation using Pearson's correlation coefficient in this section, it is important to acknowledge its limitations, including problems of existence in certain cases, restriction to linear associations between variables, and lack of invariance under monotonic transformations (Salvadori et al., 2007; De Michele et al., 2005; Serinaldi et al., 2022). To address these limitations, this approach can be extended to Spearman's rank correlation coefficient, which offers advantages in handling non-linear relationships, maintains invariance under monotonic transformations, and provides more robust estimations when dealing with outliers or non-normal distributions. The complete derivation of the geometric interpretation using Spearman's correlation is presented in Appendix A.*"

2) We have added Appendix A, which provides a complete derivation of the geometric interpretation using Spearman's rank correlation coefficient.

*"**Appendix A: Geometric Interpretation Using Spearman's Rank Correlation***

*The geometric interpretation presented in Section 2 can be extended to Spearman's rank correlation coefficient ($\rho$), which offers several advantages over Pearson's correlation ($r$), including better handling of non-linear relationships and invariance under monotonic transformations. Here we present the complete derivation:*

*Instead of working with the original variables directly, we first transform the data into ranks and then into pseudo-observations:*

$$q^S = F_Q(Q) = \frac{rank(Q)}{N+1}, s^S = F_S(S) = \frac{rank(S)}{N+1} \tag{A1}$$

*where $q^S$ and $s^S$ are the pseudo-observations representing the probabilistic ranks of discharge and surge respectively, $F_Q$ and $F_S$ are the empirical cumulative distribution functions, $rank(Q)$ and $rank(S)$ are the ranks of observations, and $N$ is the sample size.*

*Similar to the Pearson-based approach, we can represent these transformed variables as vectors in the subject space. The length of these vectors can be calculated as:*

$$\left|\overline{q^S}\right| = \sqrt{(q_1^S)^2 + (q_2^S)^2 + (q_3^S)^2 + \cdots + (q_N^S)^2} \tag{A2}$$

*with the squared length being:*

$$\left|\overline{q^S}\right|^2 = \sum_{i=1}^{N}\left(q_i^S\right)^2 \tag{A3}$$

*The standard deviation of the transformed variables is given by:*

$$\sigma_{q^S} = \frac{\left|\overline{q^S}\right|}{\sqrt{N-1}}, \sigma_{s^S} = \frac{\left|\overline{s^S}\right|}{\sqrt{N-1}} \tag{A4}$$

*The Spearman correlation coefficient ($\rho$) can then be expressed geometrically as the cosine of the angle between the transformed vectors:*

$$\rho = cos(\theta^S) = \frac{\sum_{i=1}^{N} q_i^S s_i^S}{\sqrt{\sum_{i=1}^{N}\left(q_i^S\right)^2 \sum_{i=1}^{N}\left(s_i^S\right)^2}} \tag{A5}$$

*This formulation maintains all the geometric properties discussed in Section 2, including the relationship between the angle $\theta$ and the correlation coefficient, but offers additional robustness to non-linear relationships between the original variables Q and S. Like the Pearson-based approach, uncorrelated variables are represented by perpendicular*

*vectors (θ = 90°), while perfectly correlated variables have parallel vectors (θ = 0° or 180°).*

*The key advantage of this Spearman-based geometric interpretation is that it captures monotonic relationships between the variables, not just linear ones, making it particularly suitable for analyzing compound flooding drivers that may exhibit complex, non-linear dependencies. Additionally, the rank transformation makes the approach less sensitive to outliers and more appropriate for non-normally distributed data, which is often encountered in environmental extremes.”*

**References:**

De Michele, C., Salvadori, G., Canossi, M., Petaccia, A., and Rosso, R.: Bivariate statistical approach to check adequacy of dam spillway, Journal of Hydrologic Engineering, 10, 50-57, 2005.

Salvadori, G., De Michele, C., Kottegoda, N. T., and Rosso, R.: Extremes in nature: an approach using copulas, Springer Science & Business Media2007.

Serinaldi, F., Lombardo, F., and Kilsby, C. G.: Testing tests before testing data: an untold tale of compound events and binary dependence, Stochastic Environmental Research and Risk Assessment, 36, 1373-1395, 2022.

**C#2)** In the manuscript you have considered/referred to two variables (Q and S). If you have more than two variables, it could be interesting to say how to proceed, through a pairwise analysis?

*Answer:* Thank you for this important question about handling more than two variables. We have updated the text to address your comment and clearly mention that for more than two variables, the analysis proceeds through pairwise comparisons, with each pair being visualized in the subject space:

> “*The subject space provides an effective approach when dealing with more than two variables, e.g., multi-driver compound flooding from discharge, surge, precipitation, and wind waves. It is inherently difficult to illustrate 4-dimensional scatterplots, and the interactions of multiple flooding drivers cannot be visually captured by such a plot. In such cases, Euclidean geometry offers a systematic solution through pairwise analysis. Each pair of flood drivers can be represented as vectors in a 2-D plane, with their angular separation revealing their dependence structure. This pairwise projection approach allows for clear visualization and interpretation of relationships between multiple flood drivers, overcoming the limitations of multi-dimensional scatterplots while maintaining geometric intuition.*”

Specific issues

**C#3)** Lines 84-87: I suggest to report also the p-value of the correlation coefficients to show the statistical significance.

*Answer*: In the revised version, we have added *p*-values in parenthesis:

> "… *the linear Pearson's r correlation coefficient is found to be 0.96 (p-value=0.000), while the non-linear Spearman's ρ correlation coefficient is 0.84 (p-value=0.000).*" and

> "… *with Pearson's r and Spearman's ρ being 0.41 (p-value=0.005) and 0.52 (p-value=0.000), …*"

**C#4)** In eq.(9) it is missing a parenthesis "("

*Answer*: Resolved!

**C#5)** Line 121 clarify the acronym "CCF".

*Answer*: It refers to Coastal Compound Flooding. Title 3.1 has now been updated to "Application of the Angles method for visualizing Coastal Compound Flooding (CCF) dependencies."

**C#6)** Lines 153-154 "From Figure 4, it is clear that the correlation coefficient of the period 1997-2022 is greater than that of 1972-1996 since θ is smaller (thus, the cosine is greater)." I suggest also here to calculate the statistical significance of the estimates of the coefficient, also in light of the non-stationarities claim made by the authors (lines 163-164).

*Answer*: Thank you for this important point about statistical significance. We have calculated the correlation coefficients and their corresponding p-values for both periods:

- 1972-1996: Pearson's r = 0.21 (p-value = 0.393)

- 1997-2022: Pearson's r = 0.42 (p-value = 0.035)

These results align with our visual interpretation from Figure 4, showing a higher correlation coefficient in the more recent period (1997-2022). While the correlation in the earlier period (1972-1996) is statistically insignificant (p-value>0.05) that indicates the absence of sufficient evidence to reject the null hypothesis of "no correlation", the more recent period shows a statistically significant correlation. We have updated the manuscript text to include these statistical details:

> "*From Figure 4, we observe that the correlation coefficient of the period 1997-2022 (r = 0.42, p-value = 0.035) is higher than that of 1972-1996 (r = 0.21, p-value = 0.393),*

*which is reflected in the smaller angle θ (thus, the larger cosine) in the more recent period.*"

**C#7)** In Figure 8, it is not clear if all the correlations are significant. It is important to clarify which are the significant ones.

*Answer*: The updated Figure 8 now clearly distinguishes between significant and insignificant correlations. Cross marks and underlines have been added to indicate correlations with p-values > 0.05, helping readers easily identify which relationships lack statistical significance. This distinction is also explicitly stated in the figure caption for clarity.

---

## Author Comment (AC3)

Comment

The submitted manuscript „The value of visualization in improving compound flood hazard communication: A new perspective through a Euclidean Geometry lens" proposes as stated in the title: a new visualization method that aims at better representation of bivariate relationships between variables relevant for flood hazard. Unfortunately, I don't feel that this goal was achieved.

As already noted by the other reviewer, the copula approach has been used for years in visualizing compound hazards. As literature on copulas shows, e.g. https://npg.copernicus.org/articles/15/761/2008/npg-15-761-2008.pdf , probability density function plotted on a graph where marginal distributions transformed to standard normal clearly shows the strength of the correlation (Fig. 3 therein). Further, it can show the structure of the dependency (Fig. 4 therein), which enables visually to detect tail dependence, which can be more relevant to compound hazard probability of occurrence that overall correlation. The authors' approach focuses only one dimension of the compound problem – not necessary the most important.

Even if one would not like to use the copula approach due to higher complexity (though simply showing data transformed to ranks or standard normal can already be very revealing), I don't see how presenting the correlations in Fig. 4 as angles improves this compared to e.g. showing three correlations as a bar graph. At least for me, it is not intuitive right away that a large angle with represent a low correlation. In the authors' survey results, there is clearly lower understanding of correlation in the angles method compared to simply having the numerical value. The authors only compare their approach with a raw scatterplot of particular composure of points. Scatterplots can be enhanced, as noted above, to improve the visibility of correlation.

I will skip any detailed comments, as I am not convinced at all by the authors' whole approach, which doesn't contribute to any improvement in visualization of compound hazards.

*Answer*: Thank you for taking the time to review our manuscript. We appreciate your feedback, though we respectfully think that some aspects of our work may have been misinterpreted. We would like to address your concerns and clarify the purpose and contributions of our manuscript.

**A) Purpose and Contribution of the Paper**

Our manuscript focuses primarily on compound hazard communication and visualization rather than proposing a statistically superior method to existing approaches like copulas. The Angles method is introduced as a complementary visualization tool specifically designed to communicate evolving dependencies in compound flood hazards, particularly to diverse stakeholder groups including *non-technical audiences*.

It is important to emphasize that the Angles method is not a technique to model the bivariate distribution of the two flood drivers, as copulas do, but merely an alternative visualization tool of the bivariate dependence. Therefore, it is not meant to replace copula modeling, but rather to serve as an extra first visual check of the dependence before modeling the bivariate probability (with copulas or other techniques).

**B) Key Distinctions from Copula Approaches**

While we acknowledge the value of copulas in modeling compound hazards (as referenced in the works you cited, such as Scholzel and Friederichs (2008), our approach serves a different purpose:

1. **Accessibility for Non-Technical Audiences**: Copulas, while powerful, are mathematically complex and can be challenging for non-specialists to interpret. The Angles method provides a more intuitive visual representation that can be more accessible to diverse stakeholders.

2. **Temporal Evolution Visualization**: A key contribution of our work (particularly evident in Figures 4 and 5) is the ability to visualize how dependencies between flood drivers evolve over time. This aspect is crucial for communicating non-stationarity, which is an increasingly important concept in the context of climate change impacts on flood hazards.

3. **Survey-Based Empirical Evaluation**: Unlike most methodological papers, we empirically evaluated the effectiveness of our visualization approach through a survey with 91 respondents from diverse backgrounds and locations, providing evidence-based assessment of its utility in compound hazard communication.

**C) Clarification Regarding Survey Results**

You mentioned: "In the authors' survey results, there is clearly lower understanding of correlation in the angles method compared to simply having the numerical value." We would like to clarify this interpretation.

Our survey shows that for academic respondents, the overall understanding levels (those who at least slightly agreed) were identical for both numerical values and the Angles method, with 68.2% at least slightly agreeing with the understandability of each approach. For non-academics, the results do show lower immediate understanding, which is expected with the introduction of any new method.

However, when specifically examining the communication of non-stationarity (evolving dependencies), the Angles method showed significant improvements:

- For communicating to a potential audience, the Angles method enhanced the level of "agreeing or strongly agreeing" from 11.3% to 34.1% for academics, and from 4.3% to 23.4% for non-academics, compared to scatterplots.

These results indicate that while there is an initial learning curve, the Angles method offers substantial benefits for communicating the specific concept of evolving dependencies.

**D) Methodological Considerations**

**D.1) Benchmark Comparison and Methodology Justification**

Our methodological approach involved carefully designed sequential comparisons to evaluate different aspects of visualization effectiveness:

1. *Benchmark with Numerical Values*: We first compared the Angles method with numerical correlation values as a baseline benchmark. This was necessary to establish whether the new visual representation maintained the interpretability of the underlying mathematical concept. While numerical values showed higher immediate understanding among experts, they fundamentally cannot represent temporal evolution or non-stationarity without additional visualization.

2. *Comparison with Scatterplots for Non-Stationarity*: For communicating evolving dependencies (non-stationarity), numerical values alone are completely inadequate, as they can only represent static relationships at discrete time points. This limitation necessitated our comparison with scatterplots, which are the current standard in the field for visualizing bivariate relationships.

3. *Comprehensive Evaluation Framework*: Our approach follows established evaluation frameworks in visualization science that recognize different visualization methods serve different communication purposes (Munzner, 2014; Borgo et al., 2013). While numerical representations excel at precision, geometric visualizations like the Angles method excel at communicating patterns and trends, particularly temporal evolution.

It's critical to understand that simply retaining numerical methods would fundamentally fail to address the central communication challenge our paper tackles: representing evolving dependencies over time. Numerical correlation values cannot intrinsically reflect on temporal patterns without being embedded in some visual representation (e.g., time series of correlations). The Angles method specifically addresses this limitation by providing an intuitive visual framework for representing changing relationships.

Visualization has proven to be a key tool for enhancing understanding, engagement, and decision-making (Atasoy et al., 2022; Colle et al., 2023). Our survey results demonstrate that when specifically evaluating non-stationarity communication, the Angles method substantially outperformed scatterplots, with improvements in audience clarity of 22.8 percentage points for academics and 19.1 percentage points for non-academics. This provides empirical evidence that the Angles method serves its intended purpose more effectively than current standard visualization approaches for this specific communication challenge.

**D.2) Selection of Scatterplots for Comparison**

We specifically chose scatterplots for comparison because:

1. They represent the current standard practice in the compound flood hazard literature

2. They enable a direct, one-to-one comparison with the Angles method

3. They allow for assessment of both methods' capabilities in communicating the same underlying information

4. Copula-based visualizations (i.e. the joint PDF) is too complicated for non-experts to digest and connect to.

**E) Response to Other Reviewers**

We note your reference to another reviewer's comments. That reviewer suggested using Spearman's correlation coefficient instead of Pearson's correlation for the geometric interpretation, which we have addressed in our revision by:

1. Acknowledging the limitations of Pearson's correlation

2. Adding a complete derivation of the geometric interpretation using Spearman's rank correlation in a new Appendix A

3. Explaining how this alternative approach addresses issues of non-linearity and invariance under monotonic transformations

**Summary**

In summary, we believe our manuscript makes several valuable contributions:

1. Introduces a complementary visualization approach specifically designed for communicating evolving dependencies in compound flood hazards

2. Provides empirical evidence of its effectiveness through a diverse international survey

3. Addresses a critical need in compound hazard communication for tools that can bridge the gap between technical analysis and stakeholder understanding

4. Supports the broader goals of enhancing community resilience to increasing compound flood hazards

We hope this response clarifies the purpose and contributions of our manuscript. We would be happy to incorporate further improvements or clarifications in a revised version to address any remaining concerns.

In order to address your comments and to make the objectives and the contribution of this document clearer, the following amendments have been made to the text:

1. Abstract enhancement (**P1, L22-23**):

   *"This paper introduces the Angles method, based on Euclidean geometry of the so-called "subject space," as a complementary visualization approach specifically designed for*

*communicating the dependence structure of compound flooding drivers to diverse end users."*

2. Clarify communication focus (**P2&3, L60-68**):

*"Our approach aligns with established principles in visualization science that recognize different visualization methods serve distinct communication purposes (Munzner, 2014; Borgo et al., 2013). Current approaches for visualizing compound flood dependencies, including scatterplots and statistical measures, while mathematically sound, often struggle to effectively communicate evolving patterns to diverse end users. Copula-based approaches (Schoelzel and Friederichs, 2008) provide powerful statistical frameworks but can be mathematically complex for non-specialists. The Angles method complements these approaches by offering a more intuitive visual representation specifically designed for communicating temporal evolution of dependencies. This perspective is especially important for compound flood hazard communication, where conveying evolving dependencies to non-technical audiences remains challenging."*

and **P3, L84-89**:

> *"It should be emphasized that while statistical approaches like copulas provide sophisticated analytical frameworks for modeling compound flood hazards (Schoelzel and Friederichs, 2008), our focus is specifically on developing intuitive visualization techniques for effective risk communication across diverse stakeholder groups. The Angles method is not meant to replace statistical methods like copula modeling, but rather to complement them by serving as an accessible first visual check of dependency relationships for broader audiences, including non-technical end users, before proceeding with more complex bivariate probability modeling."*

3. Clarify purpose of angles method (**P4, L108-111**):
   *"While the Angles method does not capture the full complexity of dependence structures (such as tail dependencies) that copula approaches can model, its primary strength lies in its visual intuitiveness for communicating evolving dependencies. Therefore, the method presented here is primarily designed as a communication tool rather than a statistical modeling technique."*

4. Add a new subsection, "**2.2. Survey Design and Implementation**" to draw attention of the authors to one of main contributions of the paper, as per Editor's comment. Updated text in this section (**P7, L155-168**):
   *"Our evaluation framework follows established principles in visualization science (Munzner, 2014) that recognize the importance of assessing visualization techniques based on their intended purpose and audience. We designed a sequential comparative evaluation to assess both immediate understanding and effectiveness for communicating*

*specific concepts like non-stationarity. This approach allows for a comprehensive assessment of the method's strengths and limitations across different stakeholder groups. The survey consisted of questions designed to gauge the respondents' familiarity with CCF dependencies, the clarity of non-stationarity concepts, and the effectiveness of the Angles method in communicating compound hazard. Likert scale questions were used to capture the degree of agreement or disagreement on various aspects of the Angles method, including its understandability, applicability, and perceived usefulness in CCF hazard communication. The responses are subsequently grouped into two categories: academic and non-academic respondents. This classification is used to evaluate the differing perceptions of the proposed Angles method between these two groups. Academic respondents primarily included researchers, faculty, and students from various universities, while non-academic respondents comprised professionals from the industry, government agencies, and NGOs. This segmentation allows us to explore how familiarity, relevance, and clarity of the Angles method differed across these distinct sectors."*

5. Introducing this method as a complementary approach (**P17, 368-371**):
*"This study evaluated the Angles method as a complementary visualization approach specifically designed for communicating evolving dependencies in CF hazards. Rather than replacing sophisticated statistical methods like copulas, the Angles method serves a distinct purpose in making temporal patterns of dependence more accessible to diverse stakeholder groups, particularly those without technical backgrounds."*

**References:**

- Atasoy, G., Ertaymaz, U., Dikmen, I., & Talat Birgonul, M. (2022). Empowering risk communication: Use of visualizations to describe project risks. Journal of Construction Engineering and Management, 148, 04022015.

- Borgo, R., Kehrer, J., Chung, D. H., Maguire, E., Laramee, R. S., Hauser, H., Ward, M., & Chen, M. (2013). Glyph-based Visualization: Foundations, Design Guidelines, Techniques and Applications. Eurographics State of the Art Reports, 39-63.

- Colle, B. A., Hathaway, J. R., Bojsza, E. J., Moses, J. M., Sanders, S. J., Rowan, K. E., Hils, A. L., Duesterhoeft, E. C., Boorboor, S., & Kaufman, A. E. (2023). Risk Perception and Preparation for Storm Surge Flooding: A Virtual Workshop with Visualization and Stakeholder Interaction. Bulletin of the American Meteorological Society, 104, E1232-E1240.

- Munzner, T. (2014). Visualization analysis and design. CRC press.

- Schoelzel, C., & Friederichs, P. (2008). Multivariate non-normally distributed random variables in climate research–introduction to the copula approach. Nonlinear Processes in Geophysics, 15(5), 761-772.